

# Bacterial community characteristics and enzyme activities in *Imperata cylindrica* litter as phytoremediation progresses in a copper tailings dam

Tong Jia*, Tingyan Guo* and Baofeng Chai

Shanxi Key Laboratory of Ecological Restoration on Loess Plateau, Institute of Loess Plateau,
Shanxi University, Taiyuan, China
* These authors contributed equally to this work.

## ABSTRACT

This study analyzed *Imperata cylindrica* litter to determine variation in bacterial community composition and function along with enzyme activity as phytoremediation progresses. We found significant differences in physical and chemical properties of soil and litter in the different sub-dams investigated. The Actinobacteria, Gammaproteobacteria and Alphaproteobacteria were the dominant bacteria found in the litter of the different sub-dams. The alpha diversity ($\alpha$-diversity) of litter bacterial community increased over as phytoremediation progressed, while total soil carbon and total litter carbon content were positively correlated to bacterial $\alpha$-diversity. Total litter carbon and total nitrogen were the key factors that influenced bacterial community structure. Heavy metal can influence the degradation of litters by altering the composition of the microbial community. Furthermore, bacterial communities encoded with alpha-amylase ($\alpha$-amylase) dominated during the initial phytoremediation stage; however, bacterial communities encoded with hemicellulase and peroxidase gradually dominated as phytoremediation progressed. Findings from this study provide a basis for exploring litter decomposition mechanisms in degraded ecosystems, which is critically important to understand the circulation of substances in copper tailings dams.

## INTRODUCTION

Although mineral resources are the foundation of economic development, the long-term exploitation of these resources have caused a series of ecological problems that have in some cases led to the degradation of the ecological environment (*Guo & Wang, 2013*). The Northern Copper Mine in the Zhongtiao Mountains of Yuanqu County, Shanxi Province, is one of the seven largest copper mining regions in China (*Jia et al., 2018b*). This copper mine produces over seven million tons of tailings each year. Long-term mining activities in this region have destroyed its native vegetation (*Jia et al., 2017*) and have led to severe soil erosion and a serious decline in soil fertility (*Jia, Wang & Chai, 2019*), which in turn has critically degraded local ecosystems. However, previous studies have found

Corresponding author
Tong Jia, jiatong@sxu.edu.cn

that some plant species, such as *Bothriochloa ischaemum* and *Imperata cylindrica* (*Jia et al., 2018a*), colonize copper tailing dams. Of these two species, *I. cylindrica* is dominant in the study region, which produces a large amount of litter at the end of the growing season. *I. cylindrica* is the perennial herb, and placed in the family Graminae and Genus *Imperata*. Their blossom period is from April to June. It is a hardy species, tolerant to shade, high salinity and drought. *I. cylindrica* is a vigorous, creeping perennial grass with long stolons and rooting at nodes. This grass can adapt easily to a wide range of soils (*Paz-Alberto et al., 2007*). Litter is the link between vegetation and soil interface, and it plays an important role in the function of aboveground and belowground ecosystems (*Bani et al., 2018*; *Tan et al., 2020*). Litter decomposition controls both the material and chemical cycles of terrestrial ecosystems (*Cotrufo et al., 2015*; *Xu et al., 2020*). Therefore, exploring litter decomposition characteristics and associated influencing factors in a copper tailing dam will help provide new insight into understanding the crucial role that litter decomposition plays in nutrient cycling in this region.

Litter decomposition in terrestrial ecosystems is mainly controlled by litter quality, soil properties and biological factors at a local scale (*Berg, 2014*; *Fissore et al., 2016*; *Keiluweit et al., 2015*; *Santschi et al., 2018*). Among these biological factors, microorganisms are widely known to be the main driving force behind litter decomposition processes (*Bani et al., 2018*; *Zhao, Xing & Wu, 2017*). Previous studies determined that fungi are the main decomposers that produce lignin-modifying enzymes (LMEs) and cellulase (*Boer et al., 2005*; *Yu et al., 2017*). Recent studies have also shown that soil bacteria can produce cellulase, indicating that bacteria play an important role in litter decomposition processes (*Lopez-Mondejar et al., 2016*). Additionally, *Zhang et al. (2019)* found that litter bacteria encoded with beta-glucosidase (β-glucosidase) genes may improve the capacity of litter decomposition in coniferous forests. Furthermore, many studies have shown that the relative abundance of litter bacteria increases during later litter decomposition stages, and these bacteria play a crucial role in litter decomposition (*Berg, 2014*). Therefore, it is scientifically warranted to explore litter bacteria communities and the litter decomposition characteristics, particularly in areas suffering from heavy metal pollution. However, most previous relevant studies on litter were conducted in natural ecosystems. Accordingly, very little is known about litter decomposition mechanisms in degraded copper tailings dam ecosystems.

Microbial extracellular enzymatic activities have garnered much attention, and this is due to the roles they play in litter decomposition (*Schimel, Becerra & Blankinship, 2017*; *Wang et al., 2020*). According to litter substrate properties, enzymes are generally classified into cellulase, LMEs, protease and phosphatase (*Nakamura et al., 2019*; *Wang et al., 2006*). Litter decomposition is a vital process in the global terrestrial carbon cycle (*Bani et al., 2018*; *Wang et al., 2017*; *Yan et al., 2018*). Litter carbon storage is mainly composed of cellulose and lignin. Therefore, both cellulolytic enzymes and ligninolytic enzymes play a significant role in litter decomposition. Cellulolytic enzymes mainly include endoglucanase, cellobiohydrolase and *β*-glucosidase (*Fang et al., 2010*). Comparatively, the lignin component of litter is the slowest to degrade, whose

decomposition mainly depends on ligninolytic enzymes. At present, most studies on ligninolytic enzymes focus on peroxidase, laccase, polyphenol oxidase and catalase (*Johnsen & Jacobsen, 2008*; *Saiya-Cork, Sinsabaugh & Zak, 2002*). Accordingly, our study explored the relationship between bacterial communities and enzyme activities in litter, which can help us to better understand litter degradation mechanisms.

In this study, we investigate bacterial community characteristics and enzyme activities of *I. cylindrica* litter as phytoremediation progresses, while we elucidate on the main influencing factors in a copper tailings dam. We use high-throughput sequencing and chemical analysis to test the following questions: (1) whether there exist different bacterial community structures as phytoremediation progresses; (2) whether soil and litter properties affect extracellular enzymatic activities and bacterial community characteristics; and (3) whether relevant key factors vary during different phytoremediation stages.

## MATERIALS AND METHODS

### Site description and soil sampling

Construction of the Eighteen River tailings of the Northern Copper Mine (35°15′~35°17′ N, 118°38′~111°39′E) in the southern region of the Zhongtiao Mountains started in 1969. Liu Xinggang, who is chiefly responsible for Zhongtiao Mountains Non-ferrous Metal Group Limited Department of Safety and Environmental Protection, gave us verbal permission to access and sample the sub-dam near the Northern Copper Mine. This region is under the influence of a continental monsoon climate. The average annual temperature is 13.5 °C, while the annual precipitation is 631 mm (*Liu et al., 2018b*). Currently, the Eighteen River tailings dam is composed of 16 sub-dams (*Jia, Wang & Chai, 2019*). The main constituents of the dam comprise of copper tailings and artificial loess. The slope ratio of the dam is 1:6.

In April 2019, we collected samples at S516, S536 and S560 sub-dam, and the age of phytoremediation of these three sub-dams was 50, 22 and 5 years respectively. Litter on the soil surface and topsoil samples were collected in the *I. cylindrica* distribution area of each sub-dam, for which three replications were made for each sub-dam. A total of 18 litter and soil samples were collected. Samples were sealed in self-sealing plastic bags, placed inside boxes containing ice before being immediately transported to the lab. Litter samples were then subdivided into two, one being stored at −20 °C for high-throughput sequencing and the other being stored at 4 °C along with soil samples to determine physiochemical properties. The sterile gloves should be worn throughout the sampling process to avoid contamination of the samples.

### Chemical properties and enzyme activities of samples

Soil water content (SWC) was determined by means of the drying method. An elemental analyzer (vario EL/MACRO cube; Elementar, Hanau, Germany) was used to measure total carbon and nitrogen content in soil (TC_Soil and TN_Soil) and litter (TC_Litter and TN_Litter). Soil pH was measured by potentiometric method. Shaking in a soil-water (1:2.5 w/v) suspension for 30 min and then rest. Soil particle size (PS) was measured by using Mastersizer 3,000 laser diffraction particle size analysis instrument

(Malvern Co. Ltd., Malvern, UK). Soil heavy metals, which include As, Cd, Cu, Pb and Zn, were measured by Inductively Coupled Plasma-Atomic Emission Spectrometry (iCAP 6000; Thermo Fisher Scientific, Loughborough, UK). Potassium permanganate titration was used to measure catalase. 3,5-Dinitrosalicylic acid colorimetry was used to measure sucrase and cellulase, while phenol-sodium hypochlorite colorimetry was used to measure urease. Finally, iodimetry was used to measure polyphenol oxidase (Guan, 1986).

## Techniques used for DNA extraction, PCR amplification and Miseq sequencing

We initially washed nine litter samples three times in a sterile phosphate buffer solution (PBS: NaCl, KCl, $Na_2HPO_4$ and $KH_2PO_4$) before being filtered through a sterile membrane filter (0.2 μm pore size) (Millipore, Jinteng, Tianjin, China). These membrane with bacteria samples, used to extract microbial DNA, were sealed in sterile centrifuge tubes. The E.Z.N.A.® Soil DNA Kit (Omega Bio-tek, Norcross, GA, USA) was employed for the extraction of microbial plant and soil DNA under the manufacturer's protocol. The NanoDrop ND-1000 UV-Vis Spectrophotometer (NanoDrop Technologies, Wilmington, DE, USA) was used to quantify extracted DNA. Amplification of the V5–V7 hyper variable region of the 16S rRNA bacterial gene was conducted using primers 799F (5′-AACMGGATTAGATACCCKG-3′) and 1193R (5′-ACGTCATCCCCACCTTCC-3′). The PCR reactions were conducted using the following program: 3 min of denaturation at 95 °C, 27 cycles of 30 s at 95 °C, 30 s for annealing at 55 °C, and 45 s for elongation at 72 °C, and a final extension at 72 °C for 10 min. PCR reactions were performed in triplicate 20 μL mixture containing 4 μL of 5 × FastPfu Buffer, 2 μL of 2.5 mM dNTPs, 0.8 μL of each primer (5 μM), 0.4 μL of FastPfu Polymerase and 10 ng of template DNA. The resulted PCR products were extracted from a 2% agarose gel and further purified using the AxyPrep DNA Gel Extraction Kit (Axygen Biosciences, Union City, CA, USA) and quantified using QuantiFluor™-ST (Promega, Madison, WI, USA) according to the manufacturer's protocol. We conducted sequencing at Shanghai Majorbio Bio-pharm Technology (Shanghai, China), applying the MiSeq platform (Illumina, Inc., San Diego, CA, USA).

## Processing of sequencing data

We selected QIIME software (Caporaso et al., 2010) to integrate the original sequencing data of the FASTQ format. The chimeric sequences were examined and eliminated using Usearch (vsesion 7.0, http://drive5.com/usearch/). The 97% sequence similarity was identified as the operational taxonomic unit (OTU) partition threshold for the classification results and was used to calculate bacterial community diversity and relative abundance. After that, in order to obtain the classification information of the species corresponding to each OTU, each OTU sequence (97% sequence similarity) was classified and analyzed using the RDP classifier (http://rdp.cme). And the reliability threshold using the silva132/16s_bacteria database is 70%. The bacterial sequences were banked in the National Center for Biotechnology Information database under the Sequence Read Archive accession: PRJNA611544.

**Table 1** The properties of soil and litters as phytoremediation processed.

| Physical and chemical factors | | S516 | S536 | S560 |
|---|---|---|---|---|
| Soil | TC_Soil (%) | $2.143 \pm 0.788^a$ | $0.647 \pm 0.193^b$ | $0.557 \pm 0.207^b$ |
| | TN_Soil (%) | $0.143 \pm 0.0727^a$ | $0.024 \pm 0.014^b$ | $0.022 \pm 0.013^b$ |
| | C/N_Soil | $16.346 \pm 4.148^b$ | $30.198 \pm 6.656^a$ | $28.382 \pm 6.661^a$ |
| | SWC (%) | $13.055 \pm 7.513^a$ | $6.770 \pm 2.630^{ab}$ | $5.547 \pm 3.138^b$ |
| | pH | $7.944 \pm 0.252^b$ | $8.114 \pm 0.103^{ab}$ | $8.218 \pm 0.161^a$ |
| | PS (μm) | $38.367 \pm 6.030^a$ | $41.300 \pm 10.916^a$ | $36.067 \pm 7.170^a$ |
| Heavy metals | As (mg·kg$^{-1}$) | $11.937 \pm 5.475^{ab}$ | $25.441 \pm 9.495^a$ | $4.577 \pm 1.299^b$ |
| | Cd (mg·kg$^{-1}$) | $5.967 \pm 0.659^b$ | $7.580 \pm 0.833^a$ | $3.193 \pm 0.083^c$ |
| | Cu (mg·kg$^{-1}$) | $418.408 \pm 123.080^a$ | $347.032 \pm 18.937^a$ | $487.837 \pm 51.097^a$ |
| | Pb (mg·kg$^{-1}$) | $265.647 \pm 31.314^a$ | $173.073 \pm 37.910^a$ | $185.807 \pm 107.931^a$ |
| | Zn (mg·kg$^{-1}$) | $105.606 \pm 10.795^a$ | $72.359 \pm 10.873^b$ | $51.276 \pm 15.019^b$ |
| Litter | TC_Litter (%) | $43.649 \pm 0.137A$ | $43.115 \pm 0.102AB$ | $37.659 \pm 0.134B$ |
| | TN_Litter (%) | $1.269 \pm 0.041A$ | $0.748 \pm 0.238B$ | $1.172 \pm 0.072AB$ |
| | C/N_Litter | $34.442 \pm 1.177B$ | $62.567 \pm 19.759A$ | $32.240 \pm 2.048B$ |

Notes:
Abbreviations mean total nitrogen (TN), total carbon (TC), the ratio of carbon and nitrogen (C/N), soil water content (SWC) and average particle size (PS).
Data are means ± standard deviation. Significant differences between sites ($P < 0.05$) are denoted with letters (Soil and heavy metals: a > b; Litter: A > B).

## Statistical analysis

Differences in the chemical properties among soil and litter and enzyme activities of each sub-dam were tested using the non-parametric test in SPSS Statistics version 24.0. Analysis of microbial community structure was performed using SPSS Statistics version 24.0 and SigmaPlot version 14.0. Pearson's correlation coefficient was employed to analyze the relationship among the environmental factors and microbial community diversity correlation analysis, and Venn diagram were performed using R3.5.3. Non-metric multidimensional scaling (NMDS) analysis was conducted on the bacterial community structure based on Bray–Curtis Dissimilarity, and ANOSIM was used to analyze inter-group differences. Additionally, variance inflation factor (VIF) analysis was used to eliminate the high collinearity of environmental factors using the "vegan package" in R 3.5.3. Following this, redundancy analysis (RDA) was carried out in Canoco 5.0 (Microcomputer Power, Ithaca, NY, USA). Finally, We used the interactive platform Gephi to explore and visualize networks, and PICRUSt was used to predict bacterial community functions based on the KEGG database.

## RESULTS

### Sample properties and enzyme activities in *I. cylindrica* litter and soil

Soil nutrient content (TC_Soil and TN_Soil) steadily increased as phytoremediation progressed in the copper tailings dam (Table 1). The trend in SWC variation was consistent with TC_Soil and TN_Soil. C/N_Soil was highest in the S536 sub-dam. Significant differences in pH were observed between S516 and S560 sub-dams ($P < 0.05$). The heavy metals As and Cd accumulated in S536. For litter, TN_Litter and TC_Litter
**Table 2 Enzyme activities of *Imperata cylindrica* litters over different years of phytoremediation.**

| Enzyme activities | S516 | S536 | S560 |
|---|---|---|---|
| Cellulase $(mg \cdot (g \cdot 72\ h)^{-1})$ | $0.849 \pm 0.068^a$ | $0.774 \pm 0.040^a$ | $0.458 \pm 0.228^b$ |
| Urease $(mg \cdot (g \cdot 24\ h)^{-1})$ | $3.330 \pm 0.095^a$ | $3.520 \pm 0.466^a$ | $3.087 \pm 0.293^a$ |
| Sucrase $(mg \cdot (g \cdot 24\ h)^{-1})$ | $2.998 \pm 0.625^a$ | $2.638 \pm 1.649^a$ | $2.482 \pm 0.348^a$ |
| Catalase $(mg \cdot (g \cdot 20\ min)^{-1})$ | $5.271 \pm 0.701^a$ | $3.191 \pm 0.193^b$ | $1.063 \pm 0.351^c$ |
| Polyphenol oxidase $(mL \cdot g^{-1})$ | $3.200 \pm 0.447^b$ | $5.300 \pm 0.837^a$ | $3.800 \pm 0.837^b$ |

**Note:**
Data are means ± standard deviation. The different case letters indicate that the means are significantly different among different years of phytoremediation ($P < 0.05$).

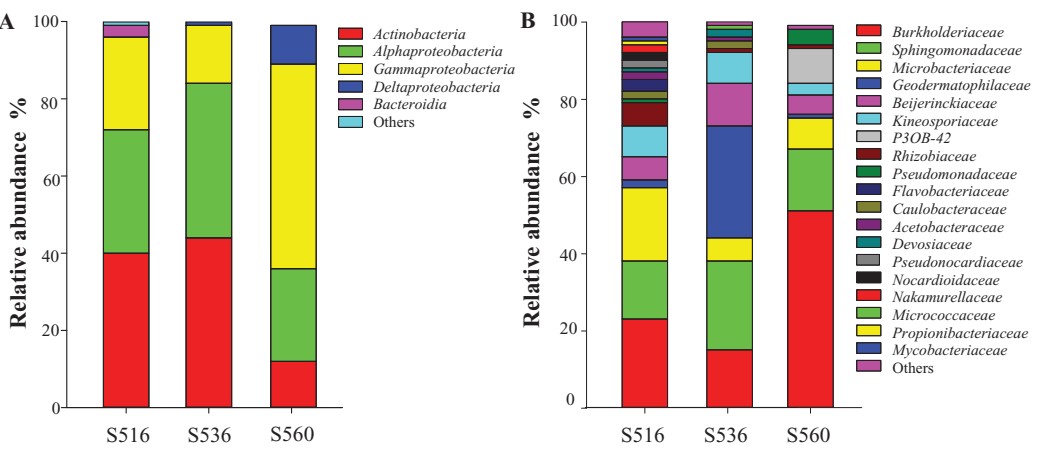

**Figure 1 The relative abundances of litter bacterial community at the levels of class (A) and family (B) with different years of phytoremediation.**

increased as phytoremediation progressed. The highest C/N_Litter value was observed in the S536 sub-dam (Table 1).

We found significant differences in cellulose, catalase and polyphenol oxidase in the litter of the three sub-dams ($P < 0.05$). Cellulase and catalase activities increased significantly as phytoremediation progressed. The highest polyphenol oxidase value was observed in the S536 sub-dam, while the lowest was observed in the S516 sub-dam. Additionally, no significant differences were found in urease and sucrase in litter among the three sub-dams (Table 2).

## Litter bacterial community composition and diversity

The litter bacterial community structure differed over phytoremediation stages. OTU numbers of the bacterial community were highest in the S516 sub-dam (i.e., 298 OTUs), followed by the S536 (i.e., 198 OTUs) and the S560 (i.e., 163 OTUs) sub-dams. A total of 162 OTUs were shared in the bacterial community litter of the S516 and S536 sub-dams. Additionally, the three sub-dams shared 119 common OTUs. Proteobacteria, Actinobacteria and Bacteroidetes were the dominant bacterial phyla in the three sub-dams (Fig. 1). Proteobacteria had the highest relative abundance of the three sub-dams, while the relative abundance of the S560 sub-dam was significantly higher compared to the

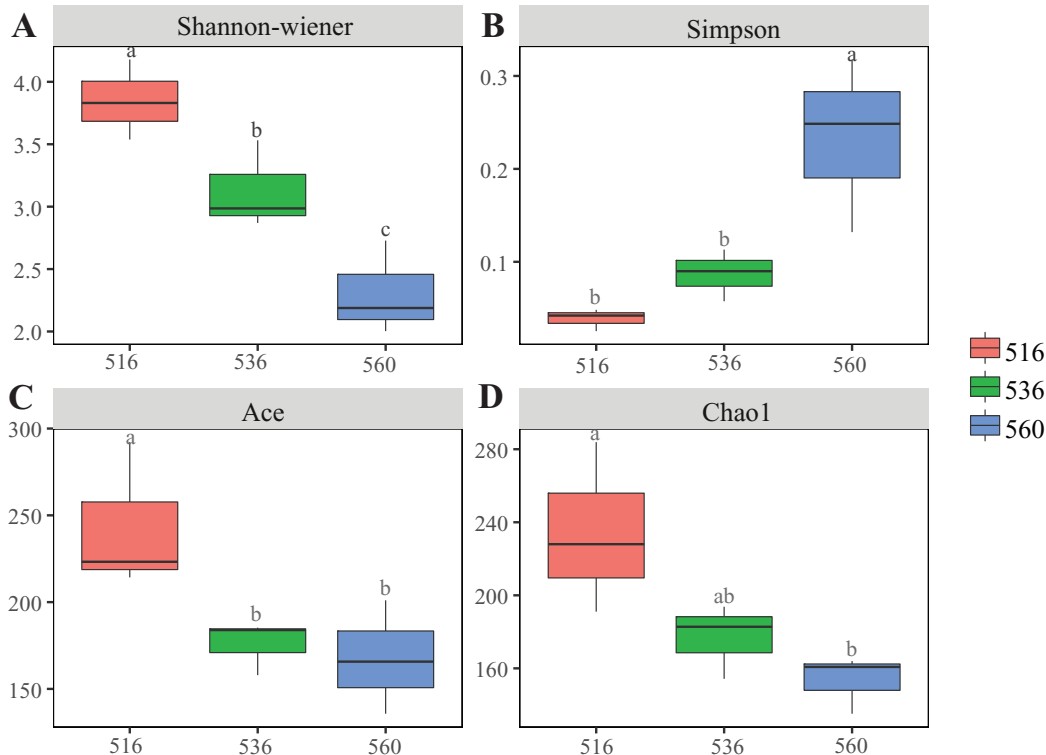

**Figure 2 Diversity indices of litter bacterial community with different years of phytoremediation.**
The bacterial diversity indices included Shannon-Wiener (A), Simpson (B) Ace index (C) and Chao1
(D). Different lowercase letters indicate significant differences, $P < 0.05$.

other three sub-dams ($P < 0.05$). The dominant bacteria in the three sub-dams were
the classes Actinobacteria, Gammaproteobacteria and Alphaproteobacteria. The relative
abundance of Gammaproteobacteria was the highest in the S560 sub-dam, and the
relative abundance of Deltaproteobacteria gradually decreased with an increase in
phytoremediation (Fig. 1A). Moreover, Burkholderiaceae and Sphingomonadaceae were
the dominant bacterial families in the three sub-dams. Burkholderiaceae was the dominant
bacteria in the S560 sub-dam, while Geodermatophilaceae abundance was highest in the
S536 sub-dam (Fig. 1B).

Estimations using the Ace and Chao1 indexes showed bacterial communities gradually
increased as phytoremediation progressed, being highest in the S516 sub-dam. Variation
trends from the Shannon index were consistent with the richness indexes (i.e., Ace and
Chao1), revealing that there were significant differences among the three sub-dams
($P < 0.05$). However, the Simpson index gradually decreased as phytoremediation
progressed (Fig. 2). NMDS analysis was performed on litter bacterial communities as
shown in Fig. 3. Given that the stress value was 0.018 in this study, NMDS analysis results
were considered well representative. Samples typically clustered together as
phytoremediation progressed, while ANOSIM showed significant differences in bacterial
community structure among the three sub-dams ($P = 0.001$).

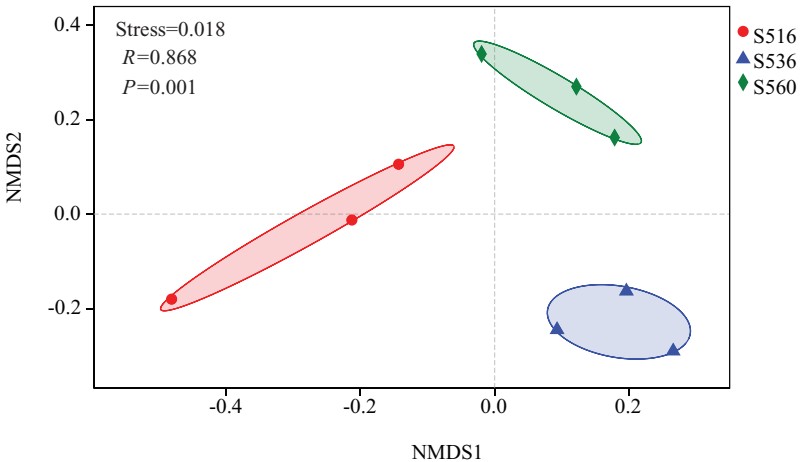

**Figure 3 Non-metric multidimensional scaling (NMDS) of litter samples at different years of phytoremediation based on the relative abundance of bacterial OTU.**

## Ecological factor effects on bacterial community litter

Correlation analysis showed a significant relationship between bacterial community litter diversity and TC_Soil, TN_Soil and pH (Fig. 4A). TC_Soil was significantly positively correlated to the Shannon-wiener index and bacterial richness indexes, but pH was negatively correlated to bacterial diversity. However, only TC_Litter was positively correlated to bacterial community diversity (Fig. 4B). Catalase was significantly positively correlated to all diversity indexes, while cellulase activity was positively correlated to the Shannon index (Fig. 4C). Zn was positively correlated to the litter community diversity and richness indexes (Fig. 4D).

Although soil and litter factors affected bacterial community structure, the main influencing factors varied in the different sub-dams (Fig. 5). We used VIF analysis to screen and remove high collinearity of physicochemical factors. RDA analysis showed that 43.22% of bacterial community variation was explained by physicochemical properties in soil. Moreover, axis 1 of the RDA plot explained nearly 26.24%, and axis 2 explained a further 16.98%. The bacterial community structure was mainly influenced by TN_Soil and SWC in the S516 sub-dam (Fig. 5A). This study also evaluated litter trait effects on bacterial community structure (Fig. 5B). Results showed that 70.5% of bacterial community variation in litter could be explained by litter traits. Both TC_Litter and TN_Litter had a significant effect on bacterial community structure (Fig. 5B). For enzyme activities, 76.65% of variation could be explained by extracellular enzyme activities (Fig. 5C). Axis 1 explained nearly 58.74%, and axis 2 explained a further 17.91% (Fig. 5C). Urease, cellulase and polyphenol oxidase activities all had a significant effect on bacterial community structure (Fig. 5C). Additionally, dominant bacteria families, such as Sphingomonadaceae, Geodermatophilaceae and Beijerinckiaceae, were positively correlated to enzyme activities in the S516 and S536 sub-dams (Fig. 5C). Furthermore, Zn and Cd respectively had a significant effect on bacterial community structure in S516

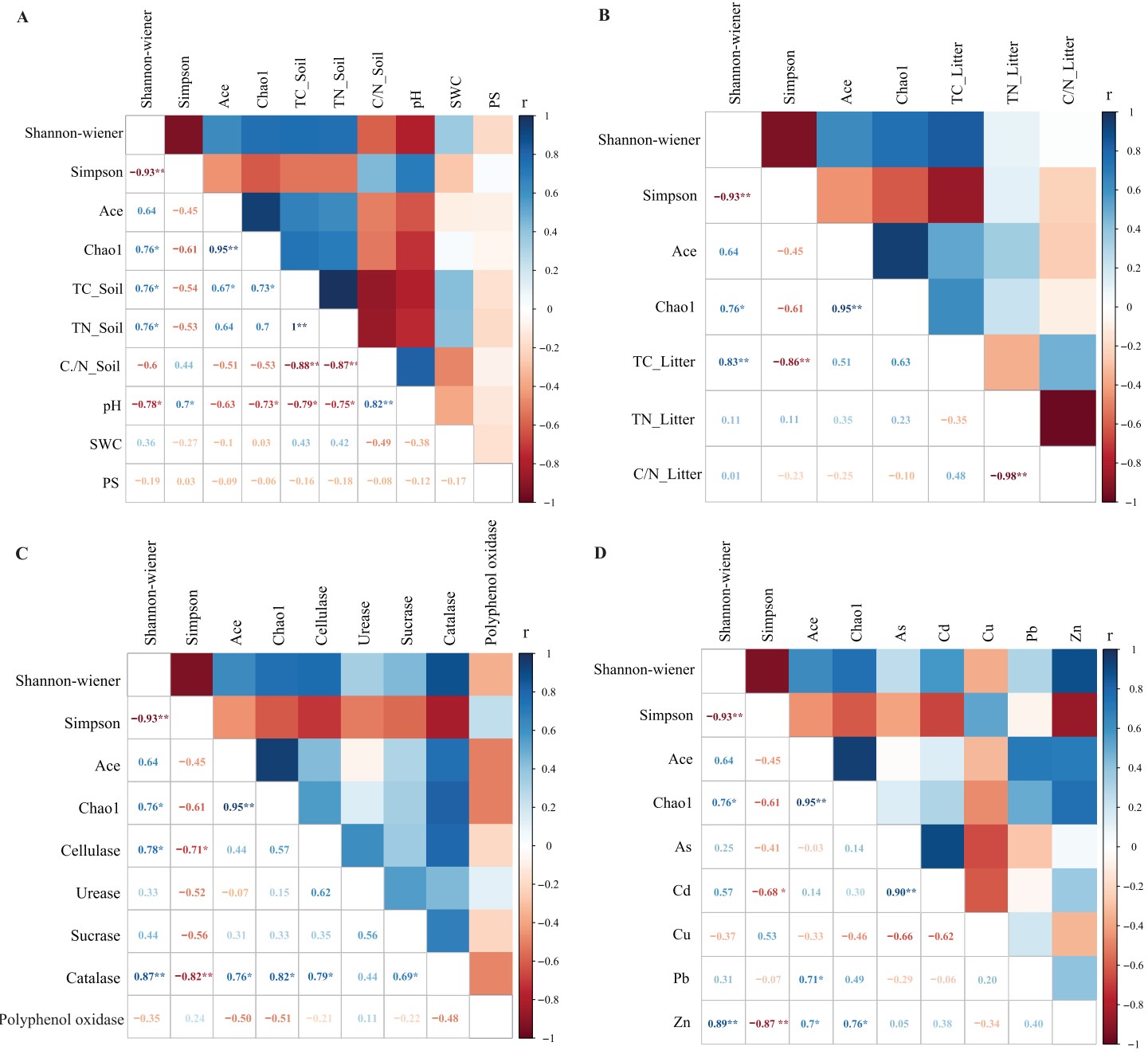

**Figure 4 The Pearson correlation between the environmental factors and litter bacterial diversitie.** The environmental factors included soil properties (A), litter properties (B), enzyme activities of litter (C) and heavy metals (D). The soil properties included total carbon (TC_Soil), total nitrogen (TN_Soil), ration of carbon and nitrogen (C/N_Soil), pH and soil water content (SWC). The litter properties included total carbon (TC_Litter), total nitrogen (TN_Litter) and ration of carbon and nitrogen (C/N_Litter) and the enzyme activities of litters included cellulase, urease, sucrase, catalase and polyphenol oxidase. *Correlation is significant at the 0.05 level (2-tailed), **correlation is significant at the 0.01 level (2-tailed).

and S536. The abundances of most bacteria classes were positively correlated with the contents of Zn and Cd while negatively corrected with Cu and Pb (Fig. 5D). This indicated that they could potentially play important roles in litter decomposition.

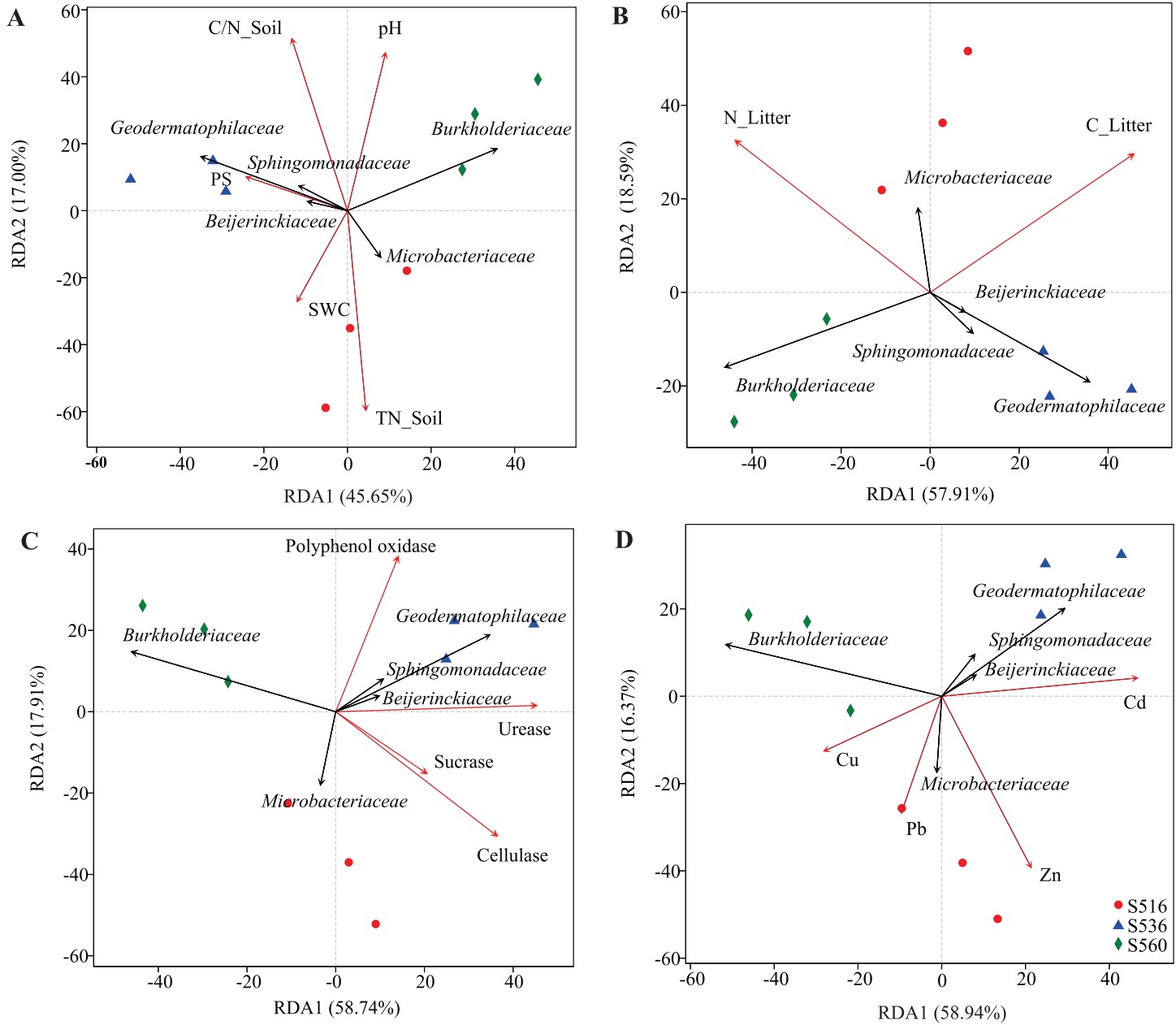

**Figure 5 RDA analysis of bacterial community and environmental factors.** The environmental factors included soil properties (A), litter properties (B), enzyme activities of litter (C) and heavy metals (D), which were represented by red arrows. The black arrows represent the bacterial which were in the top five at the family level.

## Functional characteristics of litter bacterial communities

This study identified some keystone families by building a co-occurrence network from litter bacterial communities (Fig. 6). Keystone microbes can be generally defined as those species that have a disproportionate influence on ecosystems regardless of abundance, and they are crucial in the maintenance of the stability and the function of ecosystems as well as the resistance of system disturbances. Fimbriimonadaceae was the key bacterial family in litter bacteria, and the genus *Pseudokineococcus* played critical roles in the
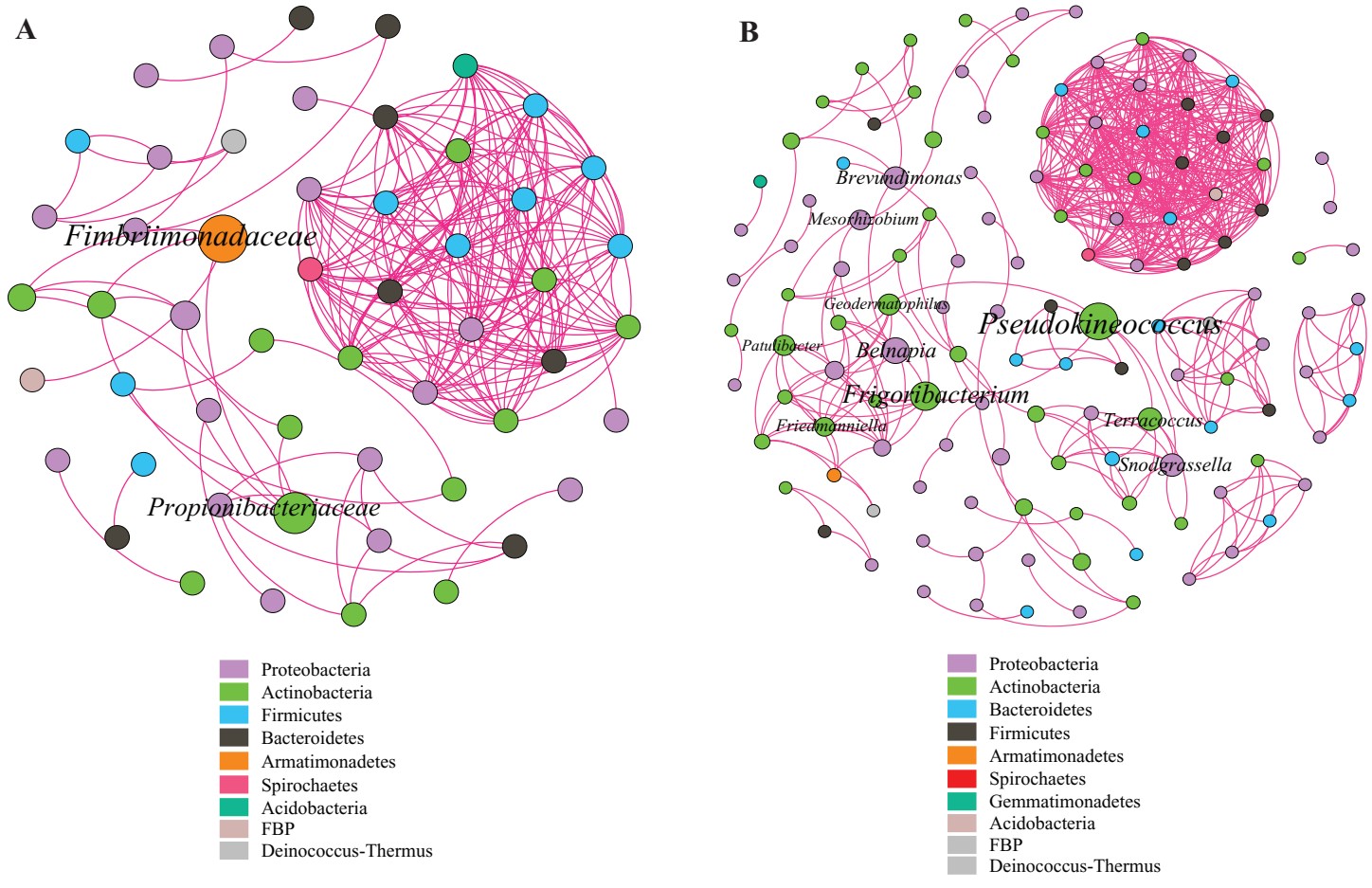

**Figure 6 Co-occurrence network of bacterial taxa on litters.** Nodes represent bacteria families (A) and genus (B), whereas pink edges, respectively, represent positive connections between pairs of species. The symbol sizes are proportional to the BC of nodes and node color represent the taxonomy of different microbial groups.

bacterial community (Fig. 6). We also used PICRUSt to infer the functional genes of bacteria associated with litter decomposition, which is based on the KEGG database. The relative abundance of genes encoded with cellulase, hemicellulase and ligninolytic enzymes notably differed among the three sub-dams ($P < 0.05$). The relative abundance of genes encoded with endo-1,4-beta-xylanase, catalase, peroxidase, endo-1,5-alpha-L-arabinosidase, alpha-glucuronidase and 1,4-beta-xylosidase were significantly higher in the S536 sub-dam compared to the S516 and S560 sub-dams. Gene encoded β-glucosidase increased as phytoremediation progressed. Additionally, only genes encoded with alpha-amylase (α-amylase) were found in amylolytic enzymes, and their relative abundance was lower (Fig. 7).

# DISCUSSION

## *Imperata cylindrica* litter and soil properties

In the study area, soil nutrient content increased as phytoremediation progressed, which is consistent with a previous study (*Jia, Wang & Chai, 2019*). The ratio between total soil

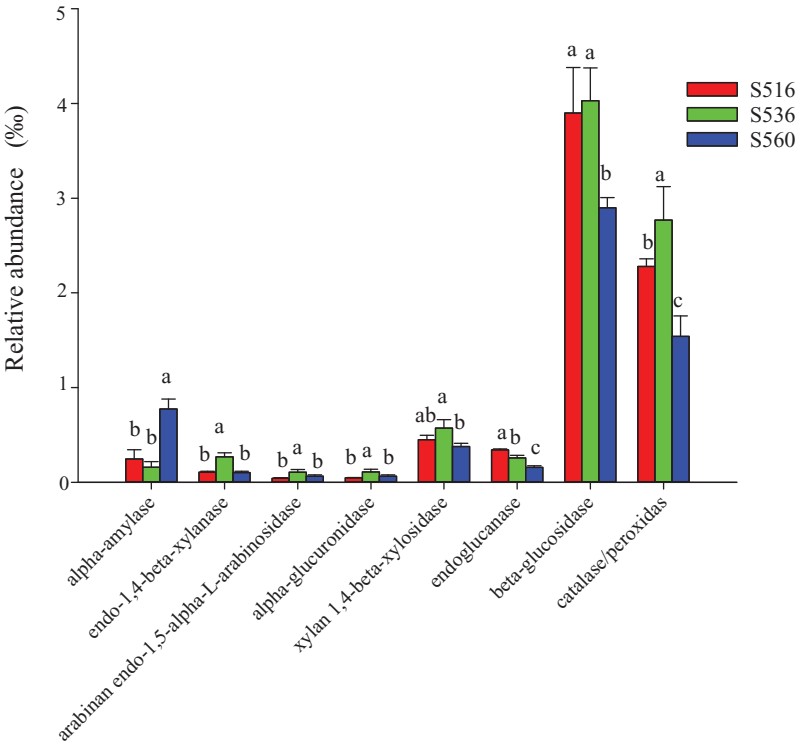

**Figure 7 The relative abundance of functional genes related to litter degradation in litter bacterial community with different years of phytoremediation.** The alpha-amylase was related to the decomposition of starch and other active substances. The endo-1,4-beta-xylanase, arabinan endo-1,5-alpha-L-arabinosidase, alpha-glucuronidase and xylan 1,4-beta-xylosidase were the enzymes related to hemicellulose decomposition. Endoglucanase and beta-glucosidase were involved in cellulose decomposition, and catalase/peroxidas were the main ligninase. Different letters indicated significant differences with years of phytoremediation, $P < 0.05$. The predicted functional analysis of the bacterial associated with litter degradation by PICRUSt.

carbon and nitrogen content reflected both the soil quality and nutrient characteristics (*Wang & Yu, 2008*), which is associated with the decomposition rate of soil organisms. Our study found higher soil C/N during different phytoremediation stages, indicating that soil TN was one of main limiting factors in the copper tailings dam. Moreover, litter C/N had an effect on litter decomposition. It has been reported that the higher the carbon is, the lower the decomposition rate will be (*Bryant et al., 1998*; *Magill et al., 2000*). In this study, the S536 sub-dam had the highest soil C/N, which could speculate that the decomposition rate of litter and soil organic carbon in this sub-dam was relatively slow compared to the other two sub-dams. Furthermore, litter decomposition is influenced by many factors. In this study, cellulase and catalase in *I. cylindrica* litter were lower during the early remediation stage when heavy metal content in soil was higher. Similarly, *Xue et al. (2018)* found that lead (Pb) inhibited cellulase and laccase activities in *Phyllostachys pubescens* litter while also inhibiting the degradation of this species. In previous studies, lead and cadmium (Cd) content seriously exceeded the copper tailings dam standard (*Jia et al., 2018a*); thus, heavy metal content in the soil of our study site could also influence litter properties.

## Bacterial community composition in *I. cylindrica* litter

Bacterial community diversity in litter gradually increased as phytoremediation progressed, which is associated with an increase in nutrient content in soil and litter (*Lu et al., 2019*). Proteobacteria and Actinobacteria were the dominant bacterial phyla in litter in all three sub-dams (i.e., S516, S536 and S560). Many studies found that Proteobacteria and Actinobacteria were the dominant bacterial phyla communities in different soil and litter types (*Bonanomi et al., 2019*; *Zhang et al., 2019*), indicating that the adaption ability of these two bacterial phyla communities to this type of environment degradation is strong, subsequently playing a vital role in litter decomposition. Moreover, Actinobacteria and Alphaproteobacteria were the dominant bacterial classes in litter in all three sub-dams, and played a critical role in carbon and nitrogen cycling (*Juhnke, Mathre & Sands, 1988*), which is essential in the decomposition of litter and the growth of plants. The relative abundance of the class Gammaproteobacteria in this study was the highest in the S560 sub-dam, which was due to the poor available nutrition during the early stage of remediation. This class, however, was strongly adaptable in its ability to dissolve phosphates in soil (*Brabcova et al., 2016*). At a family level, Sphingomonadaceae can produce cellulase, while its facility in organic matter decomposition is wide ranging, even including some complex organic matter (*Boberg, Ihrmark & Lindahl, 2011*). In our study, Sphingomonadaceae was the dominant bacterial family in all three sub-dams, indicating that its members play a critical role in litter decomposition in copper tailings dams.

## Relationships between litter bacterial communities and environmental factors

Although litter properties have been widely shown to influence bacterial community structure (*He et al., 2019*; *Yan et al., 2018*), the driving factors that affect bacterial litter communities within different ecosystems remain inconsistent between studies (*Wang et al., 2019*). Soil and litter properties affect activities associated with extracellular enzymes and bacterial community characteristics (*Petraglia et al., 2018*; *Yan et al., 2018*). In our study, total litter carbon and nitrogen significantly affected bacterial community structure. Similarly, *Xu et al. (2020)* found that carbon, phosphorus and pH were the key factors that influenced bacterial community litter and soil composition of *Robinia pseudoacacia* on the Loess Plateau. *Zhao, Xing & Wu (2017)* also found that total litter carbon was the main regulatory factor of bacterial community structure during litter decomposition processes. Moreover, *Zeng, Liu & An (2017)* found that total carbon, nitrogen and phosphorus in litter were critical factors that influenced bacterial community composition. In this study, we combined bacterial community structure analysis with extracellular enzyme activity analysis to explore the influence of environmental factors on microbial community structure. We found that urease, cellulose and polyphenol oxidase in litter were closely correlated to the bacterial community. However, critical factors varied among the different phytoremediation stages. Additionally, the families Sphingomonadaceae, Geodermatophilaceae and Beijerinckiaceae were positively correlated to a variety of extracellular enzyme activities. This was because these bacterial families are able to

produce a variety of enzymes that are used in organic matter degradation, thus playing an important role in litter decomposition processes.

Heavy metals, such as the Cu, Zn, Fe and Mg, were essential for microbial growth and participate in many important biological metabolic processed at low concentrations (*Chen et al., 2020*; *Zampieri et al., 2016*). However, at high concentrations, they had significant effect on the growth, morphology and metabolism of microorganisms, leading to a decrease in microbial diversity (*Chen et al., 2020*; *Zampieri et al., 2016*). Many studies showed that the heavy metals significantly affect the microbial abundance, diversity, the abundance of the functional classes and gene families (*Chen et al., 2018*; *Feng et al., 2018*; *Liu et al., 2018a*). In this study, the abundances of most bacteria classes were positively correlated with the Zn and Cd, indicating that these microorganisms had high tolerance to Zn and Cd. The dominant bacterial classes were positively correlated with enzyme activities associated with litter degradation. This suggested that heavy metals might influence enzyme identities and activities by altering the microbial communities compositions, then affect the litter degradations.

## Functional characteristics of litter bacterial communities

Using PICRUSt, genes encoded with cellulase, hemicellulase and LME were found in litter bacterial communities, demonstrating the critical potential of bacterial communities in litter decomposition. However, our results showed that the relative abundance of genes encoded with cellulase and hemicellulase were significantly higher in the S536 sub-dam compared to the other sub-dams ($P < 0.05$), which was inconsistent with our results on cellulase and sucrase activities. Differences between the abundance of functional genes and enzymatic activities was due to the role that other microorganisms play in litter decomposition, such as fungi. Studies found the Basidiomycota produces a wide range of LMEs and cellulase, while also controlling litter decomposition (*Zhang et al., 2019*). Furthermore, our study found that the dominate litter bacterial community could encode α-amylase at the early stage of phytoremediation (*Bani et al., 2018*). However, bacteria that encode hemicellulase and peroxidase gradually became the dominant bacterial communities as phytoremediation progressed. This could be due to the low nutrient content within the environment during the initial stage of phytoremediation, and the fact that bacteria can rapidly utilize substances in litter to meet their growth needs (*Stirling et al., 2019*). It should be noted that functional gene distribution can only predict the metabolic potential and ecological function of a bacterial community. In other words, functional gene distribution does not reflect the real metabolic activities and ecological functions of bacterial communities (*Liang et al., 2019*). Additionally, differences between functional gene abundance and gene expression have also been reported in some studies (*Hollister et al., 2010*; *Ossola et al., 2017*). Therefore, bacterial functional characteristics in litter decomposition along with gene expression and associated regulations must be further investigated in future studies.

This study suggested that bacterial community also play a crucial role in the degradation of litters, and heavy metal influence the degradation of litters by altering the composition of the microbial community. However, one limitation of this work was that we don't

cover the different stages of litter degradation. In the process of litter degradation, the chemistry of the organic substrate continuously changes, which led to various microbial community structures (*Berg & Mcclaugherty, 2008*)). In addition, the climate change was also one of the main factors affecting litter degradation (*Liang et al., 2019*). Future studies will take into account the changes in microbial community structure and function at different degradation of the *I. cylindrica* litter. Such studies will further strengthen our understanding of the relationship between the microbial community and litter decomposition in pollution area, and is critically important to understand the circulation of substances in copper tailings dams

## CONCLUSIONS

This study found significant differences in physiochemical soil and litter properties within different sub-dams. Total carbon, cellulase and catalase in *I. cylindrica* litter increased as phytoremediation progressed. Actinobcteria, Gammaproteobacteria and Alphaproteobacteria were the dominate litter bacteria in the different sub-dams. Moreover, total litter carbon and nitrogen were the key influencing factors of bacterial community structure. The α-diversity of the litter bacterial community increased as phytoremediation progressed, and the total carbon soil content and the total litter carbon content were positively correlated to bacterial α-diversity. Finally, bacterial communities encoded with α-amylase were the dominant microbial communities during the initial phytoremediation stage; however, bacterial communities encoded with hemicellulase and peroxidase gradually became the dominant microbial communities as phytoremediation progressed.

### Funding

This study was supported by the National Natural Science Foundation of China (Grant No. 31600308), the fund for Shanxi "1331 Project," China (Ecological restoration of damaged soil system), the Shanxi Province Science Foundation for Excellent Young Scholars (Grant No. 201901D211196), the Scientific and Technological Innovation Programs of Higher Education Institutions in Shanxi (Grant No. 2019L0005), the Shanxi Province Graduate Education Innovation Project (Grant No. 2019SY029), the Shanxi Province Foundation for Returnees (Grant No. 2016-006), and the Higher Education Institution Project of Shanxi Province: Ecological Remediation of Soil Pollution Disciplines Group (Grant No. 20181401). The funders had no role in study design, data collection and analysis, decision to publish, or preparation of the manuscript.

### Grant Disclosures

The following grant information was disclosed by the authors:
National Natural Science Foundation of China: 31600308.
Fund for Shanxi "1331 Project," China (Ecological restoration of damaged soil system).
Shanxi Province Science Foundation for Excellent Young Scholars: 201901D211196.

Scientific and Technological Innovation Programs of Higher Education Institutions in Shanxi: 2019L0005.

Shanxi Province Graduate Education Innovation Project: 2019SY029.

Shanxi Province Foundation for Returnees: 2016-006.

Higher Education Institution Project of Shanxi Province: Ecological Remediation of Soil Pollution Disciplines Group: 20181401.

## Competing Interests

The authors declare that they have no competing interests.

## Author Contributions

- Tong Jia conceived and designed the experiments, authored or reviewed drafts of the paper, and approved the final draft.
- Tingyan Guo performed the experiments, analyzed the data, prepared figures and/or tables, and approved the final draft.
- Baofeng Chai conceived and designed the experiments, authored or reviewed drafts of the paper, and approved the final draft.

## Field Study Permissions

The following information was supplied relating to field study approvals (i.e., approving body and any reference numbers):

Zhongtiao Mountains Non-ferrous Metals Company owns the dam. Liu Xinggang, who is chiefly responsible for Zhongtiao Mountains Non-ferrous Metal Group Limited Department of Safety and Environmental Protection, gave us verbal permission to access and sample the sub-dam near the Northern Copper Mine.

## Data Availability

The bacterial sequences are available in the NCBI database at the Sequence Read Archive (SRA): PRJNA611544.

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
