# Peer review of "Bacterial community characteristics and enzyme activities in Imperata cylindrica litter as phytoremediation progresses in a copper tailings dam"

_PeerJ, doi:10.7717/peerj.9612_

## Round 0.1 · original submission · Minor Revisions

· Academic Editor

Minor Revisions

I am pleased to inform you that I expect that your manuscript will be acceptable for publication publish in PeerJ after minor revision for a few issues raised in the comments of the reviewers. The comments of the reviewers are enclosed, and I hope the information provided by the reviewers will help with your final revision work.

Reviewer 1 ·

Basic reporting

no comment

Experimental design

no comment

Validity of the findings

no comment

Additional comments

The manuscript entitled “Bacterial community characteristics and enzyme activities in Imperata cylindrica litter as phytoremediation progresses in a copper tailings dam” presented the comparison of bacterial community characteristics and enzyme activities of I. cylindrica litter depends on the different stage of phytoremediation progresses. The experiments were well-designed and the results were explained well with proper discussion. However, there are some important things needed to be explained and revised before its publication, therefore, I suggest it to be published with major revision.

1. The abstract should be re-written.
2. Give the details of sampling method. Generally, the surface sample could be easily affected by external factors. How did the authors collect the samples of litter and soil?
3. In the session of “Materials & Methods”, were the heavy metal elements and enzyme measure for litter? If yes, I curious what happened to these in the soil.
4. As mentioned in the manuscript, the decomposition rate will be inhibited due to the C/N ratio. However, the C/N ratio of S560 is not the lowest, but its decomposition rate is low?
5. Line 250-251: “This class, however, was strongly adaptable in its ability to dissolve phosphates in soil (Brabcova et al. 2016).” It seems the phosphate is an import factors for he microbial communities. I recommend authors add the phosphate data for the soil and litter.
6. The concentration of Cu is the highest heavy metal in the litter, but I did not see proper discussion about its impact on litter decomposition or phytoremediation.
7. The information was poor in Table 1, which could be replaced by a sentence in the manuscript. Please provide more details or deleted.
8. Line 103-104: “Soil pH was measured after shaking in a soil-water (1:2.5 m/v) suspension for 30 min.” Give detail explanation of pH measurement, m/v or w/v?
9. What is d in Fig. 5? Explain more for Fig. 6.
10. Line 246-247: “Moreover, Actinobacteria and Alphaproteobacteria were the dominant bacteria in litter in all three sub-dams.” These 2 are classes, not bacteria.
11. Check the full manuscript, some spelling mistakes and improper sentences are existing in the manuscript.

Reviewer 2 ·

Basic reporting

A field study was conducted in this study and describes about the “Bacterial community characteristics and enzyme activities in Imperata cylindrica litter as phytoremediation progresses in a copper tailings dam”
Although. each part of the explained well, some basic information need to include for the more clarity of the manuscript.

Experimental design

No comments

Validity of the findings

No comments

Additional comments

Introduction: Include the life span of Imperata cylindrical
Discuss how long its take to degrade litter of I. cylindrical in a control condition (without any contamination)
Discussion need to correlated the accumulation of metal in I. cylindrical with its degradation ability by microorganisms.
Also discuss the comparison of the results with control condition (without metal influence).

Reviewer 3 ·

Basic reporting

A few confusing lines that need to be improved in writing:
1. Line 141: the section contains both litter and soil properties, so change the title accordingly.
2. Line157-158: "A totally 43 OTUs were shared .." among two sub-dams but "the three sub-dams aired 119 common OTU"? How come the overlapped amount from three is more than that from two?
3. Line 160-161.
4. Throughout the manuscript, the authors mentioned "significant differences" many times, do they mean statistically significance? If so, please mentioned p-value as well.

Literature references:
1. Line 241: "which is consistent with a previous study" - what is consistent with the previous study? The sentences seems to convey that part of the study was covered by a previous study. Please revise. Additionally, the referenced publication can not be found.

Figures:
1. Figure 1 and 6. Why did the author report analysis on both the levels of class and families or families and genus? What additional information can be gained (please expand in the main text)?
2. Figure 2. Why did the author report four methods for bacterial diversity index? What's the difference in p-value with lowercase "a" v.s. "b"?
3. Figure 3. Please explain "stress" in the main text and why it represents "NMDS analysis results were considered well representative" (Line175).
4. Figure 4 and Line 180-181 "positively correlated to all diversity indexes" - I see negative correlation to Simpson index.

Experimental design

Methods:
Line 114-115. The authors used 0.2um filter to sterilize the sample before extracting DNA for 16s analysis. However, most bacteria can not be passed through 0.2um filter.

Line 125. Please explain more about sequence analysis and OTU classification.

Validity of the findings

no comment

Additional comments

In general, the authors had performed intensive analysis on the litter and soil properties from three different sub-dams that covers different stages of phytoremediation progress. The reporting and analysis has incredible amount of details. However, some analysis tend to be repetitive (as mentioned above). I would recommend to consolidate the repetitive analysis and focus on the big picture.

Reviewer 4 ·

Basic reporting

Manuscript # 47489
Title: Bacterial community characteristics and enzyme activities in Imperata cylindrica litter as phytoremediation progresses in a copper tailings dam.
Authors: Tong Jia, Tingyan Guo, Baofeng Chai

The manuscript is well written - clear, professional and articulate English has been used throughout. Sufficient background and rationale for the work has been included in the manuscript. The authors have used adequate statistical analysis and the results from the study exhibit merit in enhancing our understanding of the bacterial community structure in sites such as the ones used in the study.

I have the following comments about the manuscript, please see below.

1. Based on the methods described, soil samples were collected only once in April 2019. Conclusions as mentioned are therefore based on samples from just one day of sampling of 18 soil samples. I am not convinced if such strong conclusions may be drawn based on a single sampling event. While I do understand that it might not have been possible to take further samples due to several restrictions we face as researchers, this limitation should be clearly indicated in the discussion section of the manuscript.

2. More citations are required in introduction part of the manuscript. Please avoid making broad statements with only one citation to provide support. For example see below:

Line 42-43: add more citations to support this statement
Line 51-52: add more citations to support this statement
Line 64=65: add more citations to support this statement


3. Lines 34-35: Are there additional studies that has been done on this? If so, please add some more.

4. Lines 59-60: this is a run-on sentences, please rephrase this for better understanding.

5. Methods, Lines 120-121: please provide more detail on PCR cycling conditions here. This should include final concentration of primers used, concentration of template DNA used in each reaction, and cycling program temperatures including annealing temperature and number of cycles run.

6. Table 3, Enzyme activity over different years of phytoremediation: This table could be better represented as a figure.

Experimental design

Please see my full review in the basic reporting section.

Validity of the findings

Please see my full review in the basic reporting section.

---

## Round 0.2 · accepted · Accept

· Academic Editor

Accept

Your revised manuscript was significantly improved according to the reviewer's comments. So, I am pleased to inform you that your manuscript is accepted to publish in PeerJ.